# The Role of Metalloproteinases and Their Tissue Inhibitors on Ocular Diseases: Focusing on Potential Mechanisms

**DOI:** 10.3390/ijms23084256

**Published:** 2022-04-12

**Authors:** Miłosz Caban, Katarzyna Owczarek, Urszula Lewandowska

**Affiliations:** Department of Biochemistry, Medical University of Lodz, 92-215 Lodz, Poland; milosz.caban@stud.umed.lodz.pl (M.C.); katarzyna.owczarek@umed.lodz.pl (K.O.)

**Keywords:** ocular diseases, matrix metalloproteinases, tissue inhibitor of matrix, metalloproteinases, gelatinases

## Abstract

Eye diseases are associated with visual impairment, reduced quality of life, and may even lead to vision loss. The efficacy of available treatment of eye diseases is not satisfactory. The unique environment of the eye related to anatomical and physiological barriers and constraints limits the bioavailability of existing agents. In turn, complex ethiopathogenesis of ocular disorders that used drugs generally are non-disease specific and do not act causally. Therefore, there is a need for the development of a new therapeutic and preventive approach. It seems that matrix metalloproteinases (MMPs) and tissue inhibitors of metalloproteinases (TIMPs) have a significant role in the development and progression of eye diseases and could be used in the therapy of these disorders as pharmacological targets. MMPs and TIMPs play an important role in the angiogenesis, epithelial-mesenchymal transition, cell invasion, and migration, which occur in ocular diseases. In this review, we aim to describe the participation of MMPs and TIMPs in the eye diseases, such as age-related macular degeneration, cataract, diabetic retinopathy, dry eye syndrome, glaucoma, and ocular cancers, posterior capsule opacification focusing on potential mechanisms.

## 1. Introduction

The ocular surface is composed of various tissue components including a specialized stratified epithelium that acts as an effective barrier to environmental, microbial, and inflammatory injuries [1]. All these components work together to maintain the integrity and function of the ocular surface. Unfortunately, due to aging, allergies, and chronic diseases such as diabetes, cardiovascular disease, and hyperthyroidism, the global incidence of eye disease is increasing. Vision quality can be disturbed by several eye diseases, such as cataracts [2], glaucoma [3], diabetic retinopathy (DR) [4], dry eye [5], age-related macular degeneration (AMD) [6] or retinoblastoma [7], all of which can significantly lower the standard of living.

Undiagnosed or untreated DR, AMD, and cataracts remain the leading causes of sight loss in the world [8]. Although a cataract is a kind of reversible blindness and surgery is an effective way to restore vision, it may further lead to multiple postoperative complications and infections [9]. For example, the posterior lenticular capsule may become opaque during various periods after cataract surgery [10]. Therefore, new, safe pharmacological alternatives to surgical intervention are needed.

All of the aforementioned eye diseases are closely associated with extracellular matrix (ECM) remodeling, which plays an important role in both its pathogenesis and its regulatory systems [2,11,12]. It has been reported that matrix metalloproteinases (MMPs) and their tissue inhibitors (TIMPs) are actively involved in this process [13]. Our understanding of MMP biology is continually expanding as new functions of MMPs in ocular tissue are discovered. While their levels are low or insignificant in most tissues during stable conditions, changes, in particular, MMP/TIMP complexes and under the influence of other mechanisms connected with both genetic and environmental factors result in the progressive degradation or accumulation of ECM structural elements, which may finally contribute to the development of eye diseases [14]. As a result, changes in MMP expression or/and activity have been proposed as possible biomarkers for the diagnosis and prognosis of particular pathological disorders. MMP inhibitors have also been applied to reverse the effects of MMPs and to assess whether MMPs are of importance for specific biological processes or pathological conditions.

Thus, the main purpose of this review was to present the role of endogenous and exogenous inhibitors of MMPs in the development of eye diseases, and their value as biomarkers or therapeutic targets. However, it should be noted that the precise role of each MMP in eye disorders remains unclear, therapies have their limitations and the mechanisms causing their upregulation are mostly undiscovered. Hence there is a clear need to investigate the potential of new-generation biological and synthetic inhibitors with greater MMP specificity and fewer side effects. These could play an important part in targeting specific MMPs, reducing unrestrained tissue remodeling, and improving the management of MMP-involved eye disorders.

## 2. Metalloproteinases (MMPs)

MMPs are zinc-dependent endopeptidases. They are among the most important proteolytic enzymes for tissue remodeling and extracellular matrix [15]. Currently, at least 23 MMPs have been investigated in humans and they are widely expressed in every tissue, including the eye. They are initially secreted as inactive pro-MMPs; these are cleaved into active MMPs by proteases or other MMPs, which interact with various protein substrates in the ECM and cell surface. They are divided into six groups based on their chemical substrate or function: collagenases, gelatinases, matrilysins, stromelysins, membrane-type (MT)-MMPs, and other MMPs [13]. They play a leading role in the modification of various ECM components, including collagen, elastin, gelatine, matrix glycoproteins, and proteoglycan degradation [13]. For example, the collagenases (MMP-1, 8, 13, and 18) degrade collagen I, II, and III, while the gelatinases (MMP-2 and MMP-9) degrade collagen I and IV. The stromelysins (MMP-3 and MMP-11) and matrylisins (MMP-7) act on elastin, fibronectin, and laminin. Finally, the MT-MMPs include MMP-14 (acting on type I, II, and III collagen), MMP-15, MMP-16, MMP-17, MMP-24, and MMP-25 [16].

MMPs are generated by connective tissues and pro-inflammatory cells such as fibroblasts, osteoblasts, endothelial cells, macrophages, neutrophils, and lymphocytes under the control of various hormones, growth factors, and cytokines [16]. Their expression and activity can be regulated via multiple cellular signaling pathways, such as mitogen-activated protein kinases (MAPKs), and by various signal transduction pathways, such as p38 mitogen-activated protein kinase (p38 MAPK), c-Jun N- terminal kinase (JNK), extracellular signal-regulated kinases (ERK), and nuclear factor-kappa B (NF-κB) [17,18]. MMPs are regulated at the gene transcription level, by post-translational modifications of the MMP protein, or by the endogenous or exogenous activation of enzyme precursors.

To maintain healthy tissue, ECM must regenerate, and replace old or damaged proteins with new ones. However, these degradation and synthesis mechanisms might be disturbed, leading to connective tissue disorders [19]. MMP proteolytic expression is tightly controlled by endogenous tissue inhibitors of metalloproteinases (TIMPs). Four members of the TIMP family have been recognized so far (TIMP-1, TIMP-2, TIMP-3, and TIMP-4) [20]. When the MMP/TIMP balance is disturbed, the resulting change in net MMP activity influences ECM turnover and tissue remodeling [15]. While MMPs play important roles in the normal physiological functions of connective tissue during the process of embryogenesis, morphogenesis, wound healing, and angiogenesis [11,21], the abnormal expression has been implicated in chronic degenerative diseases such as cardiovascular disease (atherosclerosis), musculoskeletal disorders (arthritis), neurodegenerative diseases (Alzheimer’s, Parkinson’s), diabetes or various cancers [22,23,24,25] (Figure 1). There is a growing body of evidence that the proteolytic activity of MMPs also influences eye physiology, with multiple studies reporting upregulated levels of some MMPs (MMP-1, -2, -7, -9 or MMP-13, -14) in the eye [26,27,28,29,30]. Research suggests that MMPs have a significant role in the etiology of primary open-angle glaucoma and that they may serve as molecular markers in such conditions. [3,31]. They are believed to act mainly on the remodeling of the trabecular meshwork, which is responsible for ensuring adequate aqueous humor outflow [31].

Hence, the development of MMP inhibitors that can directly suppress MMPs gene expression and/or activity provides an opportunity for creating novel pharmacological strategies. However, due to the high structural homology within the MMP family, synthetic MMP inhibitors have generally failed in vivo or in clinical trials, mainly because of their poor individual MMP selectivity, high risk of side effects, and low bioavailability [32]. Despite this, the most recent generation of MMP inhibitors has improved pharmacokinetics and required selectivity, and greatly improved toxicity profiles. Their application should provide opportunities to prevent diseases at a much earlier stage. Currently, only doxycycline, a very broad MMP inhibitor, is approved by the US Food and Drug Administration (FDA) for the treatment of disorders associated with elevated MMP activity [33].

## 3. The MMPs and TIMPs in Eye Diseases

### 3.1. Ocular Cancers

While primary intraocular cancers are relatively rare diseases, delayed diagnosis and inadequate therapy result in a poor prognosis, and the condition can be fatal if left untreated. Eye cancers, as well as carcinogenesis, have a multifactorial etiology, and a complex molecular pathogenesis that remains unclear [34]. Nevertheless, MMPs, especially the gelatinases MMP-2 and MMP-9, have a significant role in the development and progression of cancers, including those of the colon, lung, prostate, and breast. Moreover, their expression and activity are regulated by a range of signaling pathways, mainly NF-κB, Akt, phosphoinositide 3-kinase (PI3K), JNK, and ERK1/2 [35].

#### 3.1.1. Uveal Melanoma

One of the most common intraocular cancers in adults is uveal melanoma (UM), which forms from neuroectodermal melanocytes in the choroid, ciliary body, and iris. Moreover, metastases in the liver are often detected in the course of UM [34]. Its development may be strongly influenced by MMP and TIMP levels: Immunohistochemical studies have found MMP-1, -2, -9, and -19, membrane-type 1-MMP and physiological inhibitors of MMPs (TIMP-2 and -3) to have a heterogeneous distribution and intensity consistent with regional differences in the tumor microenvironment [36].

Immunostaining studies indicate that more than 20% of UM tumor cells demonstrated medium MMP-1, -9, and -19 expression while over 70% demonstrated membrane-type 1-MMP expression. Interestingly, strong MMP-2 expression was observed in tumor vasculature and stromal cells. In addition, protein expression of MMP-1, MMP-9, MT-MMP1, and TIMP-2 has been detected in the UM vasculature, and MMP-9 in the extracellular matrix, indicating that MMPs play a significant role in UM growth and angiogenesis [36]. It has also been evidenced that the high gene expression of *MMP-9* observed in UM is associated with reduced survival in the later stages of the disease [37]. Moreover, increased expression of *MMP1*, *MMP2*, *MMP9*, and *MMP16* mRNA was observed in UM; this resulted in poor overall survival, lowered disease-free survival, and intensified infiltration by immune cells, particularly Th1 cells, Th2 cells, and T follicular helper cells, indicating that MMPs are involved in UM-related inflammation and the immune response [12].

The expression of certain MMPs results in poorer prognosis; for example, increased MMP-9 expression positively correlates with metastatic profile, degree of tumor necrosis, and the formation of metastases [38,39]. In addition, downregulation of MMP-2 limits the migratory and invasive potential of UM cells; this is observed during knockdown of the leucine-rich repeat-containing G protein-coupled receptor 4 (LGR4) and reduced expression of the MMP inducer (CD147) [40,41]. In addition, *TIMP-3* gene downregulation, observed in UM tumors with monosomy of chromosome 3, is a predictive factor for disease progression and poor prognosis [42].

The molecular mechanisms of the MMP activity in UM are complex and not fully understood. However, it seems that MMP-9 activation is enhanced in UM by vascular retinal pericytes being a component of the vasculature. Additionally, UM cells appear to induce the transition of human retinal pericytes to UM-activated fibroblasts, resulting in increased expression of active MMP-9 in UM cells. It seems that the conversion of retinal pericytes to UM-activated fibroblasts is molecularly induced through binding platelet-derived growth factor (PDGF) with PDGF receptor beta (PDGFRβ) causing the activation of the signal transducer and activator of transcription 3 (STAT-3) signaling pathway and nuclear translocation of phospho-STAT3. It also increases the expression of vascular endothelial growth factor (VEGF) and transforming growth factor-beta (TGF-β) in UM-activated fibroblasts, inducing tumor progression and promoting angiogenesis in UM. It is hence clear that activated and changed retinal pericytes play a role in the progression of UM. In turn, PDGFRβ blockade limited MMP-9 expression and significantly inhibited the migration and invasion of UM cells, demonstrating the dependence of MMP-9 expression on PDGFRβ activation [43]. Further, UM cells are able to stimulate MMP-2 production in scleral fibroblasts, resulting in scleral degradation and facilitating the local invasion of UM into, and through, the sclera [44]. Furthermore, MMP-2 and MMP-9 take part in facilitating the metastasis and progression of UM, and their synthesis may be also enhanced by other mechanisms. Their expression may be also induced by epidermal growth factor (EGF), and by Grb2-associated binder 2 (Gab2) [45].

Gab2 is a member of the DOS/Gab family of scaffolding adapters. It is characterized by the presence of a C-terminal portion with numerous tyrosine phosphorylation sites, as well as a N-terminal pleckstrin homology domain and proline-rich motifs. When Gab2 is stimulated, it interacts with SH2-domain-containing proteins, thus promoting the migration and invasiveness of cancer cells. Evidence suggests that Gab2 is regulated by the PI3K pathway in cancers [45]. In addition, Gab2 facilitates the epithelial-mesenchymal transition (EMT) of cancer cells, the key process in cancer metastasis; this is mediated by enhanced MMP expression, mainly MMP-9, via activation of the mitogen-activated protein kinase (MEK) and extracellular signal-regulated kinase-1/2 (ERK1/2) signaling pathways [46].

Moreover, MMP-9 expression is promoted by the activation of the protein kinase B (Akt) and PI3K pathways, and their regulation plays a key role in exacerbating UM growth and progression. Interestingly, these two signaling pathways are regulated by the high-mobility group AT-hook1 (HMGA1) protein: this is known to modify transcriptional activity and chromatin structure in multi-genes by regulating the transcriptional factors binding to the minor groove of AT-rich DNA sequences. It is also known to play a role in UM invasiveness [47]. Downregulating the NF-κB signaling pathway also limits the migration and invasiveness of UM cells, due to lowered MMP-2, MMP-9, and VEGF expression [48]. Interestingly, UM cells also express high constitutive levels of MMP-8. This MMP degrades various types of collagen and non-collagenous ECM components and is involved in UM progression by tissue remodeling. Its expression is significantly upregulated by tumor necrosis factor-alpha (TNF-α), one of the key factors for UM aggressiveness, and this change is mediated by the phosphorylation of mitogen-activated protein kinases, such as ERK1/2 and JNK1/2 [49,50].

MMP-14 takes part in the aggressiveness of UM, whose active form is localized on the cell surface. Its exposure at the cell surface and consequent activity is regulated through the phosphorylation/dephosphorylation of the tyrosine (Thr) residues in the cytoplasmic tail of MMP14 by, inter alia, protein tyrosine phosphatase 4A3 (PTP4A3). Active MMP-14 degrades the components of the ECM such as collagen type I, II, and III, laminins 1 and 5, and fibronectin, and promotes cell migration and invasiveness in UM [51].

#### 3.1.2. Retinoblastoma

Retinoblastoma is the most frequent ocular cancer in the pediatric population. The condition develops from retinal cells. It is characterized by high proliferation and tendency to necrosis and thereby requires diagnosis at an early stage of the disease to prevent loss of sight and the eyeball, and even death during childhood [52]. Studies indicate that activated MMPs enable the migration and angiogenesis of retinoblastoma cells, and even contribute to chemoresistance. The levels of MMP-1, MMP-2, and MMP-9 have also been found to have a significant positive correlation with the intensity of retinoblastoma invasion and the occurrence of metastases [53]. In addition, higher levels of MMP-2 and MMP-9 in retinoblastoma cells are associated with an elevated risk of optic nerve invasion by cancer [54], suggesting that they may be responsible for increased malignancy of retinoblastoma, and their presence indicates the need for more aggressive therapy. Higher MMP-9 expression is associated with a more advanced stage of disease; MMP-2 is believed to regulate the differentiation of retinoblastoma cells via the activation of ERK1/2 pathway [53,55]. Furthermore, MMP-9 and VEGF expression are positively correlated in retinoblastoma tissue [53], possibly due to the fact that MMP-9 plays a key role in the acquisition of an angiogenic cell phenotype by supporting VEGF secretion [53] while VEGF promotes MMP-9 expression through the activation of ERK pathway [56].

Furthermore, MMP-3 and MMP-13 mRNA and protein expression may be induced by homeobox gene B 5 (*HOXB5*), which is upregulated in retinoblastoma cells; this promotes cell migration and invasiveness, which are mediated through stimulating the ERK1/2 pathway [57]. In addition, enhanced TIMP-1 and TIMP-2 expression are sometimes observed in metastatic retinoblastoma, and this can increase invasiveness. TIMPs are well known to inhibit MMP activity and their production is generally decreased in cancer cells; however, they are also implicated in the activation of MMPs, and may thus promote retinoblastoma progression. Therefore, the interaction between MMPs and TIMPs may have an important role in the progression of retinoblastoma [58]. Interestingly, Reinhard and co-workers indicate that the lowered mRNA expression of *MMP-2* is associated with poorer prognosis and chemotherapy resistance in retinoblastoma; however, no such change was observed at the protein level of pro- or active-MMP-2. Moreover, declined gene expression of *TIMP-2* also contributed to metastasis and therapy resistance [59]. As such, the interplay between TIMPs and MMPs in retinoblastoma seems to be a complex one that requires further investigation.

A number of mechanisms are responsible for the enhanced expression of MMP-2 and MMP-9 in retinoblastoma, and thus the promotion of proliferation, invasion, migration of retinoblastoma cells: these include Wnt/β-catenin, NF-κB, PI3K, and Akt pathway activation, the upregulation of *HMGA2* gene expression, as well as elevated expression of suppressor of Zeste 12 homolog (SUZ12), a component of the polycomb group protein [60,61,62,63,64]. Nevertheless, a decline in MMP production and activity results in significant inhibition of retinoblastoma development and progression [65,66,67,68]. Interestingly, the selective inhibition of MMP-2 and MMP-9 in retinoblastoma reduces the production of TGF-β1, the key factor for EMT, known to facilitate invasion and metastasis [66] providing an insight into the wider role of MMPs.

### 3.2. Age-Related Macular Degeneration

Age-related macular degeneration (AMD) is the main cause of irreversible blindness among patients older than 65 years in developed countries. The condition is characterized by loss of function and degradation of retinal pigment epithelial cells and photoreceptors, together with pathological matrix remodeling and new blood vessel formation; this has been attributed to the inflammatory response and oxidative stress [69,70,71]. The most recent data indicate that dysregulation of the extracellular matrix function, and its regulatory system based on MMPs and TIMPs, play an important role in the pathogenesis of both dry and wet AMD [72,73]. The conditions are characterized by changes in the level of gelatinases: patients with AMD demonstrate elevated levels of MMP-2 in the retinal pigment epithelium-associated interphotoreceptor matrix and MMP-9 in plasma [74,75]. However, other studies indicate no significant increase in serum levels of MMP-1, MMP-2, MMP-3, MMP-9, TIMP-1, and TIMP-3 in patients with AMD compared to those without [76].

It is widely known that aging is a risk factor for AMD. One of the explanations may be the fact that the MMPs level increases with age in Bruch’s membrane, where initial pathological changes occur in AMD, predisposing to its disruption by proteolytic processes [73]. In addition, one of the early symptoms of AMD is the occurrence of drusen, small extracellular deposits of various molecules and proteins. These are also observed under retinal pigment epithelial cells in Bruch’s membrane during oxidant injury to the retina, characterized by a decrease in MMP-2 activity. Hence, it is possible that MMP-2 may play a key role in the early stage of the disease. Interestingly, MMP-14 overexpression is believed to prevent retinal oxidative injury in AMD by maintaining the appropriate activity of MMP-2 [77,78,79]. MMP-14 also plays a key role in extracellular matrix degradation at focal adhesions in human retinal pigment epithelial cells [80]. Furthermore, MMP-14 may be involved in the process of MMP-2 activation, which occurs when the level of TIMP-2 is reduced in the relation to MMP-14 in the pro-MMP-2/TIMP-2/MMP-14 complex [78].

Further, MMP-1 and MMP-3 are also believed to play a role in the pathogenesis of AMD. Elevated protein levels of these molecules were observed in retinal pigment epithelial cells exposed to oxidative stress. Their activity may be mediated by the ERK1/2 and p38 MAPK pathways. A shift in the MMP-1,-3/TIMP-1 ratio was found to contribute to the intensive degradation of type I collagen. It seems that this change may have an important role in the initiation of early exudative AMD [81].

AMD manifests as dry and wet forms, dependent on the formation of pathological blood vessels. The wet form, i.e., with neovascularisation, has been characterized by the presence of MMP-2 and MMP-9 in the retinal pigment epithelium-Bruch’s membrane, vessels, and stroma; indeed, MMP-9 is known to promote an angiogenic phenotype among choroidal endothelial cells in AMD [82,83]. Interestingly, MMP-9 may take part in pathological vessel formation in two ways: by inducing extracellular matrix degradation, facilitating choroidal neovascularization, and stimulating endostatin, an endogenous inhibitor of angiogenesis [82]. Nevertheless, these data require further analysis. A positive correlation has been found between the MMP-9 level in the vitreous humor and the quantity of subretinal fluid in patients with wet AMD, suggesting that MMP-9 may be a potential biomarker of subretinal exudate [84]. They found that mice with laser-induced choroidal neovascularization with a lowered expression of MMP-9 also had decreased volume of laser-induced choroidal change [85]. In addition, the in vitro expression of MMP-9 was found to be dependent on TNF; this plays a key role in the development of neovascular AMD, and is governed by the JNK pathway [85,86]. Additionally, there is evidence that VEGF could promote choroidal neovascularization, an element of neovascular AMD, by stimulating the secretion of MMP-2 and MMP-9 in retinal pigment epithelial cells [87]. It is likely that the changes occurring during choroidal neovascularization are governed by a feedback loop between VEGF and MMPs; hence, the proteins of the extracellular matrix may increase the secretion of VEGF by retinal pigment epithelial cells. Further, VEGF is able to upregulate the expression of MMPs in the retina, with hypoxia being one of the triggers [87,88,89].

Interestingly, a recent study indicated that MMP-2 may influence AMD by modulating the complement system. Fernandez-Godino and co-workers report the presence of a disorganized fiber network of collagen IV among human fetal retinal pigment epithelial cells cultivated on Bruch’s membrane from subjects with AMD compared to those without AMD. They also note an elevation in MMP-2 activity and C3a levels in the AMD patients. This result may indicate a relationship between abnormal extracellular matrix production, MMP-2 level, and the complement system in AMD [90]. In addition, angiotensin II, a molecule taking part in the regulation of hypertension being a risk factor of AMD, may affect the activity of MMPs in AMD. Elevated angiotensin II level was found to enhance MMP-2 activity in human retinal pigment epithelial cells in vitro; this was accompanied by elevated levels of MMP-14, intensified type IV collagen degradation, and greater ERK and p38 MAPK pathway stimulation [91,92].

A literature review found that TIMPs take part in the development of AMD. High levels of TIMP-3 are related to lower levels of ECM components in the Bruch’s membrane and lower ECM thickness in AMD [93,94]. Furthermore, AMD patients tend to demonstrate differences in MMP-9, TIMP-1, and TIMP-3 plasma concentrations. In addition, plasma levels of TIMP-1 and MMP-9 proteins are elevated in patients with geographic atrophy, one of the late stages of AMD, while TIMP-3 and TIMP-3/MMP-2 ratios are lowered in subjects with AMD with choroidal neovascularization [95].

### 3.3. Diabetic Retinopathy

Diabetic retinopathy (DR) is a form of microangiopathy that affects retinal blood vessels, arising as a complication of diabetes mellitus. It remains one of the leading causes of sight loss worldwide. Its prevalence is expected to increase to over 190 million cases by the year 2030. There are two main types of DR: non-proliferative and proliferative associated with the formation of pathological new vessels and vitreous hemorrhage [96]. MMPs and TIMPs are believed to be associated with vascular complications in patients with diabetes mellitus and DR. For example, the severity of DR is significantly associated with a higher level of MMP-2 in plasma [97]. Hyperglycaemia associated with DR causes a number of biochemical, structural, and functional changes in the retina and vasculature [4]. An increase in MMP-2 protein expression and MMP-9 activity was demonstrated in glucose-induced rhesus macaque choroid-retinal endothelial cells and bovine retinal endothelial cells, and these effects seem to be mediated by the activation of the Akt and ERK pathways [98,99]. Additionally, MMP-2 and MMP-9 upregulation was observed in the retinal pigment epithelium in streptozocin-stimulated Sprague-Dawley rat DR models in vivo [100,101]. The action of MMPs in DR is complex. A number of MMPs, especially MMP-2, MMP-9, and MMP-14, take part in the breakdown of the blood–retinal barrier (BRB) by degrading proteins, mainly occludin and cadherin, responsible for maintaining the BRB integrity; this results in increased vascular permeability and BRB disruption [102,103].

The breakdown of the BRB is associated with the occurrence of diabetic macular edema, a critical hallmark of DR [104]. Patients with diabetic macular edema are characterized by higher levels of MMP-1 and MMP-9 in the aqueous humor compared to patients without, suggesting that these two enzymes promote the prevalence of diabetic macular edema [105]. Further, MMPs may regulate the inflammatory response in DR. Increased levels are observed in diabetes mellitus, and MMPs are known to cleave the monocyte chemoattractant protein that regulates the inflammatory condition [106]. In addition, MMPs participate in the recruitment and diapedesis of leukocytes occurring in the retina during DR [107]; MMPs, especially MMP-2 and MMP-9, are also responsible for ECM component degradation, a key process of angiogenesis [108]. In addition, in DR, hypoxia is observed in the retinal tissue; this contributes to the upregulation of hypoxia-inducible factor 1 alpha (HIF-1α) expression, promoting the production of VEGF [109]. In turn, VEGF stimulates the expression of MMPs, mainly MMP-2, as observed in Müller cells [110]. Additionally, MMP-9 facilitates the apoptosis of retinal Müller cells by mitochondrial damage, increased pro-apoptotic Bax protein expression, and decreased anti-apoptotic Bcl-2 protein expression, resulting in the progression of DR [111].

Clinically, increased levels of MMP-1, MMP-7, MMP-9, MMP-14, TIMP-1, and TIMP-4 were noted in human vitreous. Interestingly, the levels of MMP-1, MMP-9, MMP-14, TIMP-1, or TIMP-4 were found to significantly correlate with VEGF level, an angiogenic factor promoting the development of proliferative DR, in the vitreous fluid, indicating that MMPs modulate the angiogenesis and promote the progression of DR [29,112,113]. Additionally, higher concentrations of TIMP-1 were detected in the aqueous humor of the subjects with advanced non-proliferative and proliferative DR compared to the control subjects, or those with mild or moderate non-proliferative DR, indicating that TIMP-1 may play an important role in later stages of the disease [114]. TIMP-1 and -4 are both MMP inhibitors. Elevated TMIP-1 levels enhance angiogenesis and VEGF expression, while the increase in TIMP-4, the inhibitor of MMP-9 and controller of angiogenesis, occurs in response to angiogenic activity and elevation of TIMP-1 and MMP-9 [113].

Interestingly, TIMP-3 had strong anti-inflammatory and anti-angiogenic properties in DR. An intravitreal injection of recombinant TIMP-3 in streptozocin-induced Sprague-Dawley rats contributed to an attenuation of BRB breakdown by downregulating the protein expression of NF-κB p65 subunit and VEGF in the retinas. Additionally, TIMP-3 significantly decreased the protein expression of VEGF and phospho-ERK1/2 in high glucose-induced human retinal Müller glial cells, and reduced migration, proliferation, and chemotaxis of human retinal microvascular endothelial cells stimulated by VEGF [115].

### 3.4. Cataract and Posterior Capsule Opacification

Cataracts are a major cause of blindness and visual impairment worldwide [116]. The lens of the eye becomes clouded, resulting in visual impairment consisting of blurry vision, vision problems at night, halos around light, and impaired visual acuity, and their occurrence and extent depend on the localization and range of changes in the lens. It should be emphasized that if untreated, cataracts can lead to blindness. Cataracts can be inborn, age-related, traumatic, metabolic, and toxic in origin [117], and can be subcapsular, nuclear, or cortical depending on localization in the lens [118]. Recent studies indicate that MMPs can also participate in the pathogenesis of cataracts [119,120,121]. One study of MMP-9 found the highest expression in patients with cortical cataracts (12.14 ng/µg proteins) compared to subcapsular, (5.53 ng/µg) and nuclear cataracts (4.52 ng/µg). MMP-9 activity was determined in lens epithelial cells (LECs), obtained during phacoemulsification from patients with cataracts, by succinylated-gelatin assay. Importantly, the mean activity of MMP-9 increased with age, peaking in patients over 60 years of age. The detailed cause of the above-described dependences is not fully known. However, the occurrence of diabetes mellitus, vascular diseases, prolonged UV exposure to sunlight, and UV radiation, i.e., the risk factors of cortical cataract, increase with aging; they are associated with oxidative stress and may enhance the activity of MMP-9 [119].

It is possible that MMPs may also have a potential role in posterior subcapsular cataracts, as indicated by their activity in both serum and LECs in patients with glucocorticosteroid-induced posterior subcapsular cataracts. Higher activities of MMP-9 and MMP-2 in both serum and LECs were observed in comparison to the patients with non-glucocorticosteroid-induced posterior subcapsular cataracts. It indicates the potential significance of gelatinases in steroid-induced cataracts. In addition, the mRNA expression of *MMP-2* and *MMP-9* was in LECs over 121 times higher for MMP-2 and 274 times higher for MMP-9 in patients with glucocorticosteroid-induced cataracts than in those with non-glucocorticosteroid induced cataracts [122]. Additionally, MMPs may play an important role in anterior subcapsular cataracts. Rats with TGF-β-induced anterior subcapsular cataracts demonstrated elevated protein expression of MMP-2 and MMP-9 in lenses, and increased activity in conditioned medium from cultured lenses, indicating they may play a role in the formation of an ocular disorder. Moreover, the use of a broad-spectrum MMP inhibitor (GM6001) and an MMP-2/9 specific inhibitor suppressed the development of cloudy plaques in lenses. This may result from the inhibition of epithelial-mesenchymal transition of LECs, an important process in subcapsular cataracts [123]. During epithelial-mesenchymal transition, epithelial cells transform into motile mesenchymal cells, consequently, the epithelium loses polarity and specialized cell–cell contacts. In addition, changes in cell behavior, differentiation, and survival occur, resulting from inter alia the downregulation of epithelial markers e.g., E-cadherin, and the upregulation of mesenchymal markers, MMPs, or cell migration [124,125].

Dwivedi and co-researchers report that MMP-inhibitor treatment attenuated the mRNA expression of the mesenchymal marker alpha-smooth muscle actin (α-SMA), and the upregulation of the gene expression of E-cadherin, an epithelial marker, in rats with TGF-β-induced anterior subcapsular cataract [123]. Further, in a mouse model of anterior subcapsular cataracts, MMP-9 played a significant role in mediating the epithelial-mesenchymal transition of LECs in anterior subcapsular cataracts. However, MMP-2 expression was not found to be necessary for this process [126]. MMP-9 also appears to play a role in diabetic cataracts. In streptozotocin-stimulated rats, demonstrated elevated protein expression of MMP-9 in LECs compared to controls, and the effect was mediated by increased expression of TGF-β. This suggests that MMP-9 may affect the occurrence and development of diabetic cataracts [121]. In addition, MMP-2 has been found to demonstrate increased immunoreactivity in LECs of patients with cataracts and diabetes mellitus; MMP-2 may hence play role in the pathogenesis of cataracts in subjects with diabetes mellitus [127].

TIMPs inhibit the activity of MMPs and protect the ECM from excessive degradation. Their inhibitory action is associated with blocking access to the MMPs catalytic space [128]. Increased TIMP levels were observed in the aqueous humor of patients with cataracts, possibly to compensate for the elevated level of MMP-2 [129]. It may thus be suggested that the disorders in the synthesis and the regulation of the activity of TIMPs can increase the risk of cataracts.

Posterior capsular opacification (PCO) is an ophthalmic disorder that is a late complication of cataract surgery. Its mechanism is complex. Residual LECs after surgical removal of cataracts proliferate, undergo EMT stimulated by TGF-β, migrate from the equatorial region of the lens capsule, and express extracellular components, such as α-SMA, fibronectin, and type I collagen [130]. These changes seem to be accompanied by an increase in MMPs. For example, type I collagen, whose production is elevated in PCO, promotes the expression and activity of MMP-2 and MMP-9 [131]. In addition, MMP-2 activity was elevated in the aqueous humor of the eyes of New Zealand White rabbits following cataract surgery [132]. Furthermore, the inhibition of MMP-2 and -9 production was shown to reduce the migration of LECs, indicating that posterior capsular opacification can be limited by MMP regulation [131]. It has been observed that downregulation of MMP-2 and -9 by proteasome inhibition decreased cell migration in TGF-β2-induced LECs (HLE-B3), suggesting that gelatinase inhibition may prevent PCO [133]. These effects, including MMPs action, are mediated by Smad2/3, ERK/MAPK, and Wnt/β-catenin signaling pathways, and their downregulation results in the limitation of MMP expression and the development and progression of PCO [134,135,136].

### 3.5. Dry Eye Syndrome

Dry eye syndrome (DES) is a multifactorial eye disorder of the ocular surface caused by a loss of tear film homeostasis; it is generally, caused by disturbed tear production, excessive evaporation, or altered composition by inducing tear hyperosmolarity [137]. This, in turn, triggers the inflammatory response, damages the ocular surface, and causes neurosensory abnormalities by activating some signaling pathways, mainly NF-κB, JNK, and MAPK; this initiates the transcription of genes encoding inflammatory MMPs [138]. A positive correlation has been found between greater osmolarity and the expression of MMP-1, MMP-3, MMP-9, and MMP-13 in human corneal epithelial cells [138]. Nevertheless, MMP-9 is considered to play a key role in the response to hyperosmolar stress in DES [139,140]; it influences the remodeling of the damaged corneal surface and digests elements of the corneal epithelial barrier, resulting in its disruption [141]. An increase in MMP-9 expression and activity in the cornea is associated with mainly occludin lysis in the apical corneal epithelium [142]. Additionally, a disturbed MMP: TIMP ratio was observed in the eyes with tear dysfunction caused by hyperosmolarity and inflammatory response [143].

Hyperosmotic stress elevated MMP-2 and MMP-9 mRNA expression and activity in human corneal epithelial cells in vitro [144]. In addition, several animal models of experimental DES, including NOD.B10.H2b mice exposed to desiccation stress, Wistar rats with excised lacrimal glands, and New Zealand white rabbits simulated by concanavalin A found that MMPs had a pathological role in DES. In these studies, DES induction resulted in enhanced MMP-2 and MMP-9 expression and inflammatory response, as well as reduced tear production, corneal epithelium damage, irregularities, and detachment [145,146,147,148].

Generally, clinical studies support the use of MMP-9 as a biomarker of DES, and elevated concentrations in tears are directly correlated with the diagnosis of the disease. Detection of elevated MMP-9 levels may contribute to earlier diagnosis, more adequate therapy, and better management of DES [149,150]. Furthermore, a direct association has been found between tear concentrations of MMP-9 and tear osmolarity, and an inverse concentration between MMP-9 level and Schirmer’s test value, measuring tear production [151]. Interestingly, a combination of tear osmolarity assessment and MMP-9 detection in tears may be helpful for determining the severity of Sjögren’s Syndrome-related dry eye, a special form of DES affecting the lacrimal glands, occurring in the course of autoimmune disease [152]. Further, MMP-9 activity in tears was found to have a direct correlation with conjunctival corneal fluorescein staining, sign severity, and topographic surface regularity index in patients with DES. In addition, high MMP-9 activity was related to the reduced tear break-up time (TBUT) and visual acuity scores [153,154].

The positive association between MMP-9 activity in tears and the severity of DES results from the fact that MMP-9 is a late-stage sign and is rarely overexpressed in mild-type DES [155]. Interestingly, increased MMP-2 and MMP-9 activity has been demonstrated in the tears of postmenopausal women with DES. It is widely known that hormonal changes in women occurring during menopause contribute to a higher prevalence of DES. Moreover, it is considered that 17β-estradiol is responsible for upregulating the production of MMP-2 and MMP-9 in lacrimal glands and the conjunctival epithelium, with a consequently raised activity in tears [156]. In addition, MMP-9 and MMP-2 may impede the healing of the corneal epithelium [157]. Therefore, it is desirable to inhibit MMPs in treating DES.

### 3.6. Glaucoma

Glaucoma is a progressive ocular disorder leading to irreversible blindness, and its global prevalence in adults aged 40 to 80 years is assessed to be 3.5%. Generally, it has an asymptomatic course, and its diagnosis is frequently delayed. It is characterized by neurodegeneration of the optic nerve and the loss of retinal ganglion cells. There are two main types of the disease: open-angle and closed-angle glaucoma. Although they have different pathogeneses, both have intraocular pressure (IOP) as the major modifiable risk factor for disease progression. IOP is regulated by the production and outflow of aqueous humor secreted through the ciliary body. The majority of aqueous humor flows through the pupil into the anterior chamber and then drains by a trabecular meshwork, the lumen of Schlemm’s canal, into aqueous veins, and the episcleral venous system [15,158,159]. The trabecular meshwork is a porous tissue consisting of trabecular meshwork cells that secrete components of the extracellular matrix, such as type I, III, and IV collagen. Inadequate and disproportionate accumulation of ECM components impedes the outflow of aqueous humor, contributing to an elevation of IOP. In turn, an increase in IOP stimulates the expression of MMPs by trabecular meshwork cells, suggesting that MMPs may play an important role in glaucoma [160].

Patients with primary open-angle glaucoma are characterized by levels of MMP-1, MMP-2, MMP-3, MMP-9, and MMP-12 protein in aqueous humor, and by upregulated mRNA expression of *MMP-1*, *MMP-9*, and *MMP-12* in the blood (peripheral blood lymphocytes) compared to the subjects without any type of glaucoma, suggesting that these changes may be considered as a risk factor for the development of primary open-angle glaucoma. Interestingly, the elevated expression of these MMPs is associated with changes in the promoter sequences of the genes [161,162]. In addition, increased activity of MMP-9 has been detected in the patient’s tears with both open-angle and closed-angle glaucoma at the early stage of the disease. In turn, overwhelming the MMP-9 activity was observed in advanced stages. These changes may result from the decrease in viable trabecular meshwork cells, which may occur in the course of glaucoma, leading to a reduction of regulatory functions by limiting MMP-9 expression in advanced disease [163]. An in vivo study in a mouse model indicates that MMP-9 takes part in regulating IOP, and animals with deficiency of MMP-9 expression have elevated IOP [164].

Changes in the expression and activity of MMPs occurring in glaucoma may have a significant influence on the course of the disease. It is widely known that alterations ongoing in the extracellular matrix affect aqueous humor outflow and modulate the development of glaucoma [165]. MMP-9, whose expression and activity in glaucoma are mainly upregulated during the early stages of the disease, helps maintain the ultrastructural organization of the trabecular meshwork. This enzyme digests IV-type collagen, laminin, and fibronectin; their incorporation into the trabecular meshwork is disrupted in glaucoma, facilitating the outflow of aqueous humor and decreasing IOP [166,167]. Moreover, it has been demonstrated that the MMP-9 null mice had aberrant collagen composition of the trabecular meshwork, as well as lowered aqueous humor turnover and ocular hypertension, indicating that MMP-9 may be an important remodeler of trabecular meshwork mitigating the course of glaucoma [166].

MMP-2, known as gelatinase-A, may play a similar role as MMP-9. However, the level of MMP-2 in aqueous humor in patients with glaucoma remains unclear. Higher levels were revealed in the aqueous humor of subjects with acute primary angle-closure [168] but significantly lower levels were noted in aqueous samples in subjects with primary open-angle glaucoma compared to control cataract patients, with no change in TIMP-2 levels [169]. Generally, MMP upregulation is accompanied by a reduction in TIMP level. In addition, the TIMP-2 level was also found to be elevated in the aqueous humor of patients with primary open-angle glaucoma compared to the cataract controls [170]. Interestingly, these findings may indicate that MMP-2 expression is not independent, and imbalances in the MMP-2/TIMP-2 ratio may be significant in the pathogenesis of ocular hypertension in glaucoma [171]. For example, a lower MMP-2/TIMP-2 ratio in the aqueous humor is a risk factor for trabeculectomy failure in patients with acute primary angle-closure [172]. On the other hand, elevated concentrations of TIMP-4, a known inhibitor of MMP-9, have been confirmed in the aqueous humor of the patients with primary open-angle glaucoma, which would also disrupt the MMP/TIMP molar ratio [173,174]. Currently, there is no clear finding on whether the increase in TIMPs is caused by altered production of MMPs or vice versa, and the causal relationship with glaucoma.

Another MMP that seems to play a significant role in the outflow of aqueous humor and glaucoma is MMP-1. Viral vector-mediated delivery of MMP-1 has been found to reverse elevated IOP in the trabecular meshwork of sheep with steroid-induced glaucoma [175,176]. This effect could result from the fact that the MMP-1 is expressed in tissues associated with the unconventional (uveoscleral) outflow pathway in normal human eyes, such as ciliary muscle, iris root, and sclera. The MMP-1 in the sclera is likely responsible for the remodeling of scleral ECM-enhancing uveoscleral turnover [177]; this would account for the observed upregulation of MMP-1 expression in glaucoma.

It is also important to note that MMP-3 activity was found to be lower in the aqueous humor in patients with glaucoma compared to age-matched ocular normotensive controls [178]. MMP-3, known also as stromelysin-1, has a broad spectrum of proteolytic activity, and it has a number of substrates present in the trabecular meshwork, including the juxtacanalicular tissue, the outer layer of the trabecular meshwork: *inter alia* type IV collagen, elastin, fibronectin, and laminin. Moreover, MMP-3 is able to activate other MMPs, such as MMP-1 and MMP-9, which facilitate the turnover of aqueous humor [165,179,180,181]. Therefore, the lowered MMP-3 expression and activity may cause the progression of glaucoma by the impediment of aqueous humor outflow. Furthermore, intracameral inoculation of AAV-2/9 containing a CMV-driven MMP-3 gene into wild-type (C57BL/6) mice elevated the aqueous concentration and activity of MMP-3 secreted from the corneal endothelium; interestingly, this elevated MMP-3 expression enhanced outflow facility and reduced IOP, suggesting that MMP-3 plays a significant role in mitigating glaucoma [178].

As described above, the elevations of MMPs expression and activity in the aqueous humor in glaucoma prevent disease progression, as indicated by changes occurring in steroid-induced glaucoma. Long-term corticosteroid treatment is associated with structural changes in the trabecular meshwork, particularly enhanced extracellular matrix deposition in the juxtacanalicular region [182]. In addition, corticosteroids treatment has been associated with a reduction of the activity of stromelysins, type IV collagenases, and tissue plasminogen activators in trabecular meshwork organ and human corneoscleral explant cultures containing both ciliary body and trabecular meshwork [183,184]. Furthermore, intracameral gene therapy with a vector carrying an inducible MMP-1 human gene prevented further IOP increase in sheep with corticosteroid-elevated IOP [175].

One form of glaucoma treatment involves the modulation of the MMP expression and activity; prostaglandin analogs, such as latanoprost, bimatoprost, unoprostone, lower IOP by enhancing uveoscleral outflow. It is believed this takes place by increasing the protein expression of MMP-1, MMP-9, and TIMP-4 in the trabecular meshwork. In addition, bimatoprost and latanoprost are known to enhance MMP-3 and TIMP-2 protein expression, as well as MMP-1 and MMP-9 activity, facilitating aqueous humor outflow. The observed increase in TIMP-2 and TIMP-4 after therapy probably occurs as a response to an elevation in MMP level; however, the rise in TIMP level would increase the effectiveness of individual prostaglandin analogs [185].

The above data clearly indicate that the elevated MMP concentration and activity observed in the aqueous humor of patients with glaucoma has a protective role. However, studies suggest that MMPs may also have a harmful effect on glaucoma. Glaucoma is a progressive neurodegeneration of the optic nerve, which is accompanied by loss of the retinal ganglion cells caused by elevated IOP [158]. Zalewska et al., (2016) indicated that MMP-9 may be responsible for the damage to the optic nerve and retinal ganglion cells. Significant MMP-9 overexpression has been demonstrated in retinal ganglion cells, the internal nuclear layer of the retina, and glial cells surrounding optic nerve axons in absolute angle-closure glaucoma. In addition, MMP-9 may promote pathological apoptosis by the activation of cell death receptors and pro-apoptotic proteins in absolute glaucoma [186]. In addition, MMP-9 expression and activity were found to be upregulated in the retinal neurons of rats injected with N-methyl-D-aspartate and glycine into the vitreous humor; this suggests that an extracellular proteolysis pathway in the retina results in the death of retinal ganglion cell through MMP-9 activation [187]. A key role in the activation of MMP in the retina in glaucoma may be played by tissue plasminogen activator (tPA) and urokinase plasminogen activator (uPA): both levels were increased in the retinas of C57BL/6 mice after injection of staurosporine into the vitreous humor, and this accompanied increased MMP-9 activity and retinal ganglion cell death [188]. It is widely known that uPA and tPA activate plasminogen to form plasmin: a protease responsible for promoting the death of retinal ganglion cells via tissue remodeling or MMP activation [189,190,191]. It is also possible that MMP-2 may contribute to retinal ganglion cell death: MMP-2 null mice were found to demonstrate improved retinal ganglion cell survival after intravitreal injection of N-methyl-D-aspartic acid, a mediator of glaucomatous neuropathy, compared to wild-type mice [192]. The mechanisms (cellular signaling pathways) associated with MMPs and TIMPs in eye diseases are presented in Figure 2.

## 4. Novel Agents Modulating the Expression and Activity of MMPs and TIMPs in Eye Diseases

Despite the current research, available eye disease therapies are still associated with side effects and remain not fully effective. As such, there is an ongoing need to develop new therapeutic options for the treatment of ocular disorders [193,194]. As noted in the third part of this paper, MMP and TIMP expression and activity could be an attractive pharmacological target and any modulation may significantly limit the development and progression of eye diseases. This chapter describes and discusses the efficacy of novel therapeutic options aimed at MMPs and TIMPs in eye diseases based on recent pre-clinical and clinical studies.

### 4.1. Natural Agents

Polyphenols, natural components of plants and their fruits, are able to regulate cellular mechanisms, and evidence indicates that they can also modulate the course of diseases [195,196]. They have also demonstrated an inhibitory effect on eye diseases [191,197,198]. The flavonoid chrysin may play a role in DR. It was found to decrease MMP-2 protein expression in glucose-stimulated rhesus macaque choroid-retinal endothelial cells and reduced the activation of Akt and ERK signaling pathways, thus limiting cell migration [99]. Further, various polyphenols, such as chebulagic acid, chebulinic acid, gallic acid, epigallocatechin-3-gallate (EGCG), and quercetin, reduced the mRNA and protein expression and activity of gelatinases in inflamed-choroidal-retinal endothelial and -retinal pigment epithelial cells, suggesting they may have therapeutic potential in AMD [199,200,201].

Pterostilbene, an analog of resveratrol and a natural compound of blueberries, diminished the mRNA expression and activity of MMP-2 and MMP-9 in hyperosmotic stress-induced primary human corneal epithelial cells [144]. In turn, eye drops containing catechin, a natural flavonoid occurring in tea, downregulated the protein expression of MMP-2 and MMP-9 in the desiccation stress-stimulated NOD.B10.H2b mice; this change was accompanied by an increase in the tear production and a decrease in corneal epithelium irregularities and desquamation [146,148]. In an animal model of UM, curcumin treatment resulted in lowered expression of MMP-2 and MMP-9 at both the mRNA and protein level in cancer cells, causing a reduction in tumor size [202]. Eriodictyol, a natural dihydroflavonoid, caused significant inhibition of migration and invasion of retinoblastoma cells; this resulted from downregulation of MMP-2 and MMP-9 protein expression and blockage of the Akt and PI3K signaling pathways [64].

Resveratrol is believed to offer various pro-health benefits. It was found to increase MMP-2 production in the aqueous humor, reduce IOP and improve the morphology of the trabecular meshwork and retina in rats with steroid-induced glaucoma [203,204]. Miyata and co-workers showed that individual polymethoxylated flavones isolated from *Kaempferia parviflora* decreased gelatinase activity and the mRNA expression of MMP-9, and increased the protein production of TIMP-2 in phorbol 12-myristate 13-acetate-stimulated LECs via suppression of p38 MAPK, JNK1/2 and ERK1/2 pathway activation [205]. This finding highlights the efficacy of flavones in the prevention of PCO.

Paeoniflorin, a monoterpene glucoside naturally occurring in the root of *Paeonia lactiflora*, is able to inhibit the inflammation of microglia within the retina and disrupt the BRB, which plays a key role in the development of DR. It has been demonstrated that paeoniflorin reduced the activity of MMP-9, and lowered the activation of p38 and NF-κB signaling pathways in the microglial cells following high glucose treatment. Additionally, this compound decreased MMP-9 activity, lowered IL-1β protein expression, and ameliorated diabetic retinal changes in streptozocin-stimulated mice [206].

Theissenolactone C, a fungal derivate extracted from Theisseno cinerea (Xylariaceae), could be used as a retinoprotectant in glaucoma. This agent lowered the protein level and activity of MMP-9 in the retinas of rats with IOP-induced ischemia/reperfusion-retinal injury. Therefore, it offers promise as an adjuvant agent in the therapy of glaucoma; however, further clinical studies are needed [207].

Zeaxanthin, a natural compound belonging to the xanthophyll subclass of the carotenoids, has been found to have strong anti-cancer properties against UM. This compound inhibited the invasion and migration of UM cells in vitro. It is believed to act by lowering MMP-2 secretion and reducing the NF-κB level in nuclear extracts from the UM cells [208].

It is also worth noting the effect of omega-3 long-chain polyunsaturated fatty acids (PUFAs), naturally occurring in fish and seafood. They are believed to exert a protective effect against oxidative stress and inflammation in retinal diseases, such as DR or AMD, mainly by inhibiting pro-inflammatory cytokine expression, and by activating signaling pathways associated with inflammation and ROS production [209,210]. In addition, it was found that a diet containing PUFAs reduced the expression of gelatinases in choroidal-retinal explants obtained from mice with experimentally-induced choroidal neovascularization via adiponectin pathway activation. These effects limited new choroidal vessel formation, suggesting that a PUFA-rich diet may inhibit AMD development [211]. Additionally, omega-3 fatty acids have been found to mitigate the clinical course of DES. The oral supplementation using re-esterified omega-3 fatty acids, or their re-esterified triglyceride form, contributed to a decline in tear osmolarity and ocular surface disease index, and an increase in TBUT and tear production in patients with DES. The changes were accompanied by a decrease in the MMP-9 level in the tear film; this could be responsible for the observed reduction in ocular surface inflammation and a general improvement of symptoms and signs [212,213].

### 4.2. Synthetic Agents

In an in vivo model of DR, oral supplementation with synthetic melatonin diminished the protein expression of MMP-9, VEGF, and inducible nitric oxide synthase in streptozotocin/nicotinamide induced rats. It revealed that melatonin has inhibiting potential on angiogenesis in DR, suggesting that this compound offers promise in prophylaxis or supportive treatment of DR [214].

Interestingly, niclosamide, a salicylanilide with antihelminthic activity, is able to inhibit the malignant phenotype of UM. Niclosamide treatment limited the invasion and migration of UM cells, decreased MMP-9 protein expression, and suppressed the NF-κB and Wnt/β-catenin signaling pathways, indicating its strong anti-cancer properties [215].

Noteworthy are new agents for the treatment of glaucoma. Sodium 4-phenylbutyrate, a salt of aromatic short-chain fatty acid, decreased IOP in mice with dexamethasone 21-acetate-induced ocular hypertension; in addition, treatment downregulated the protein expression of collagen I and fibronectin in the trabecular meshwork tissue, enhancing the aqueous humor outflow. The treatment also prevented the dexamethasone-stimulated synthesis of ECM components and endoplasmic reticulum stress markers in primary human trabecular meshwork cells and upregulated the gene expression and activity of MMP-9 in the same model, indicating that sodium 4-phenylbutyrate was able to degrade abnormal extracellular matrix accumulation in glaucoma [216].

Statins also offer promise as agents with beneficial effects for glaucoma. The synthetic statin atorvastatin suppressed the protein expression and activity of type IV collagenases in astrocytes of the optic nerve head following TGF-β2 stimulation. This was achieved by inhibiting the RhoA/ROCK signaling pathway, suggesting that statins have a protective effect against optic nerve damage in glaucoma [217].

Selected studies investigating the modulation of MMPs and TIMPs by novel agents are summarized in Table 1, Table 2 and Table 3.

## 5. Conclusions

Both MMPs and TIMPs play important roles in the pathogenesis of a number of eye diseases, including AMD, DR, UM, retinoblastoma, DES, PCO, cataract, and glaucoma, and the regulation of eye health. In addition, their expression and activity are influenced by other cellular pathways and thus play a role in the development and progression of ocular disorders. Although the exact mechanisms of MMP and TIMP activity depend on the type of disease, they are both known to influence angiogenesis, invasion, migration, epithelial-mesenchymal transition, inflammation, and apoptosis. Moreover, it seems that the ratio of MMP/TIMP levels is more important in the development of eye diseases than the individual levels of these molecules. With the evolving use of target treatment, we recommend further studies investigating the role and potential use of MMPs and TIMPs in the diagnosis, prognosis, and treatment of eye diseases. In addition, further studies are needed to identify novel, therapeutic options modulating MMP and TIMP expression and activity, and these may slow the prevalence, development, and progression of eye diseases.

## Figures and Tables

**Figure 1 ijms-23-04256-f001:**
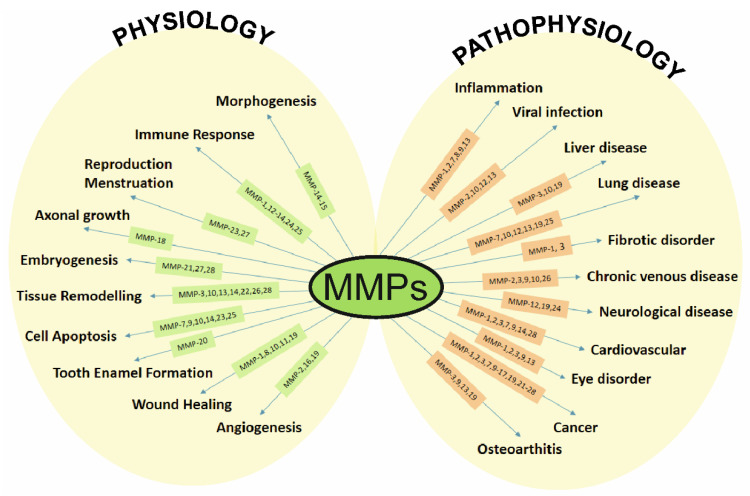
Physiological and pathophysiological processes regulated by MMPs.

**Figure 2 ijms-23-04256-f002:**
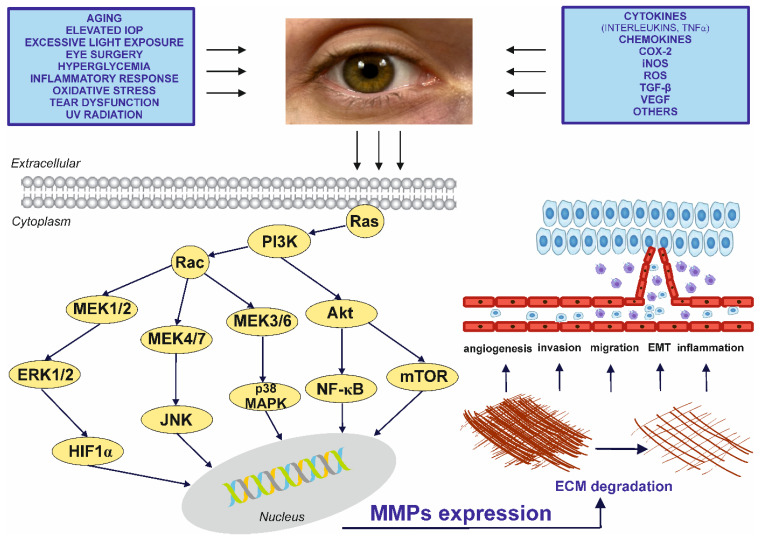
A schematic diagram showing molecular mechanisms involved in stimulation of the MMPs expression and effects being results of the MMPs action in the eye diseases. Akt—protein kinase B; COX-2—cyclooxygenase-2; ECM—extracellular matrix; EMT—epithelial-mesenchymal transition; ERK—extracellular-signal-regulated kinase; HIF1α—hypoxia-inducible factor 1 alpha; iNOS—inducible nitric oxide synthase; IOP—intraocular pressure; JNK—c-Jun N-terminal kinase; MEK—mitogen-activated protein kinase; mTOR—mechanistic target of rapamycin; NF-κB—nuclear factor-kappa B; PI3K—phosphatidylinositol 3-kinase; p38 MAPK—p38 mitogen-activated protein kinase; Ras—rat sarcoma; subfamily of small GTPases; ROS—reactive oxygen species; TGF-β—transforming growth factor-beta; TNF-α—tumor necrosis factor-alpha; UV—ultraviolet; VEGF—vascular endothelial growth factor.

**Table 1 ijms-23-04256-t001:** Overview of in vitro studies associated with the modulation of MMPs and TIMPs in eye diseases.

Agent	Cell Line	Concentration/Duration	Biological Effects/Findings	Reference
Diabetic retinopathy				
Chrysin (5,7-dihydroxyflavone)	Glucose-induced rhesus macaque choroid-retinal endothelial cells (RF/6A)	3, 10, 30 μM for 0.5, 16, 24 h	↓MMP-2 protein; ↓VEGFR1, VEGFR2 mRNA and protein; ↓VEGF, HIF1α protein; ↓p-Akt, p-ERK protein; ↓ migration	[99]
Paeoniflorin	Glucose-induced BV2 cells	0.1. 1, 10 μM for 6 h	↓MMP-9 activity; ↓p-p38 protein; ↓NF-κB translocation from the cytosol to the nucleus	[206]
Age-related macular degeneration				
Chebulagic acid	TNF-α-induced Rhesus monkey choroidal- retinal endothelial cells (RF/6A)	1, 5, 10, 25, 50, 100 μM for 4, 48 h	↓*MMP-9* mRNA, protein, activity; ↓PDGF-BB, IL-6, IL-8, MCP-1, MIP-1b, p-ERK1/2, p-NF-κB, p-p38 protein; ↓tube formation	[200]
Chebulinic acid	TNF-α-induced rhesus monkey choroidal- retinal endothelial cells (RF/6A)	1, 5, 10, 25, 50, 100 μM for 4, 48 h	↓*MMP-9* mRNA, protein, activity; ↓PDGF-BB, IL-6, IL-8, MCP-1, MIP-1b, p-ERK1/2, p-NF-κB, p-p38 protein; ↓tube formation	[200]
EGCG	H_2_O_2_/TPA/ TNF-α-induced human retinal pigment epithelial cells (ARPE-19) + VEGF-induced human retinal microvascular endothelial cells	1, 10, 25, 50 μM for 24 h	↓*MMP-9* mRNA, protein, activity; ↓MMP-2 activity; ↓VEGF, VEGFR-2 mRNA; ↓tube formation	[199]
Gallic acid (3,4,5-Trihydroxybenzoic acid)	TNF-α-induced Rhesus monkey choroidal- retinal endothelial cells (RF/6A)	1, 10, 50, 100 μM for 4, 48 h	↓*MMP-9* mRNA, protein, activity; ↓IL-6, IL-8, MCP-1, p-ERK1/2, p-NF-κB, p-p38 protein; ↓tube formation	[200]
Quercetin (3,3′,4′,5,7-pentahydroxyflavone)	TNF-α-induced human retinal pigment epithelial cells (ARPE-19)	5, 10, 50, 100 μM for 1, 4, 6, 48 h	↓*MMP-9* mRNA and activity; ↓ICAM-1 mRNA and protein; ↓p-JNK1, p-JNK2, p-ERK1, p-ERK2, p-p65 protein	[201]
Dry eye syndrome				
Pterostilbene (trans-3,5-dimethoxy-4-hydroxystilbene)	Hyperosmotic stress-induced primary human corneal epithelial cells	For mRNA expression: 5, 10, 20 μM for 4 h For protein expression and MMPs activity: 5, 10, 20 μM for 24 h	↓*MMP-2* mRNA and activity ↓*MMP-9* mRNA and activity ↓IL-1β, IL-6, TNF-α mRNA and protein ↓COX-2 mRNA and protein ↑SOD1 mRNA and protein ↑PRDX4 mRNA and protein	[144]
Ocular cancers				
Eriodictyol	Y79 cells	25, 50, 100 μM for 24 h	↓MMP-2 and MMP-9 protein; ↓ p-Akt, p-PI3K protein; ↑cleaved caspase-3 protein; ↓invasion; ↓migration	[64]
Niclosamide	92.1, Mel270, Omm1, Omm2.3 cells	1–10 μmol/L for 24, 36, 48 h	↓MMP-9 protein; ↑active caspase-3 protein; ↓NF-κB activation; ↓p-GSK3β, p-β-catenin protein; ↓invasion; ↓migration; ↓proliferation	[215]
Zeaxanthin	C918 cells	3–10 μM for 0.5, 8, 16, 24 h	↓MMP-2 protein; ↓NF-κB protein; ↓invasion; ↓migration	[208]
Posterior capsule opacification				
Flavones isolated from *Kaempferia parviflora* (5,7-dimethoxyflavone; 3,5,7-trimethoxyflavone; 3,5,7,4′-tetramethoxyflavone; 3,5,7,3′,4′-pentamethoxyflavone)	Phorbol-ester-stimulated human lens epithelial cells (SRA01/04)	0.25, 1, 4, 16, 64 μM for 24 h	↓pro-MMP-2, pro-MMP-9 activity; ↓*MMP-9* mRNA; ↑TIMP-2 protein; ↓p-JNK1, p-JNK2, p-ERK1/2, p-p38 protein	[205]
Glaucoma				
Sodium 4-phenylbutyrate	Dexamethasone-stimulated human primary trabecular meshwork cells	5 mM for 24 h/7d	↑*MMP-9* mRNA and activity; ↓fibronectin, collagen I, laminin protein; ↓GRP-94, GRP-78, CHOP protein	[216]
Statins (simvastatin, lovastatin, atorvastatin)	TGF-β2-stimulated human primary astrocytes of optic nerve head	5 μg/mL for 1 h	↓MMP-2 and MMP-9 protein; ↓MMP-2 and MMP-9 activity; ↓p-MYPT1, p-mic protein	[217]

Legend: ↑ activation or increase; ↓ inhibition or decrease; CHOP—CCAAT-enhancer-binding protein homologous protein; COX-2—cyclooxygenase 2; GRP-78—glucose regulated protein 78; GRP-94—glucose regulated protein 94; HIF1α—hypoxia inducible factor 1 alpha; ICAM-1—intercellular adhesion molecule 1; IL—interleukin; MCP-1—monocyte chemoattractant protein 1; MIP-1b—macrophage inflammatory protein 1; MMP-2—matrix metalloproteinase 2; MMP-9—matrix metalloproteinase 9; NF-κB—nuclear factor kappa B; p-Akt—phosphorylated protein kinase B; p-ERK—phosphorylated extracellular-signal regulated kinase; p-GSK3β—phosphorylated glycogen synthase kinase 3 beta; p-JNK—phosphorylated c-Jun N-terminal kinase; p-MYPT1—phosphorylated myosin phosphatase target subunit 1; p-NF-κB—phosphorylated nuclear factor kappa B; p-p38—phosphorylated p38 mitogen-activated protein kinase; p-PI3K—phosphorylated phosphatidylinositol 3-kinase; PDGF—platelet-derived growth factor; PRDX4—peroxiredoxin 4; SOD1—superoxide dismutase 1; TIMP-2—tissue inhibitor of metalloproteinases 2; TNF-α—tumor necrosis factor alpha; VEGF, vascular endothelial growth factor; VEGFR1—vascular endothelial growth factor receptor 1; VEGFR2—vascular endothelial growth factor receptor 2.

**Table 2 ijms-23-04256-t002:** Overview of in vivo studies associated with the modulation of MMPs and TIMPs in eye diseases.

Agent	Animal Model	Dose/Duration	Biological Effects/Findings	Reference
Diabetic retinopathy				
Melatonin	Streptozotocin/nicotinamide-induced Wistar rats	85 μg/d orally for 14 days	↓MMP-9 protein; ↓VEGF, iNOS protein; ↓advanced oxidation protein products	[214]
Paeoniflorin	Streptozocin-stimulated CD-1 mice	20, 40 mg/kg/d for 5 weeks	↓MMP-9 activity; ↓IL-1β protein;	[206]
Age-related macular degeneration				
Dietary Omega-3 Long-Chain Polyunsaturated Fatty Acids	Laser-induced C57BL/6J and *Apn*^−/−^ mice	Defined rodent diets with 2% omega 3-long-chain-polyunsaturated-fatty-acids (1% docosahexaenoic acid and 1% eicosapentaenoic acid) for 7 days before and after laser photocoagulation	↓*MMP-2* and *MMP-9* mRNA ↓choroidal neovessels	[211]
Dry eye syndrome				
Catechin (flavon-3-ol)	Desiccation stress-induced NOD.B10.H2b mice	1% catechin or 1% nanocomplex PEG/catechin as eye drops for 10 days	↓MMP-2 protein ↓MMP-9 protein ↓IL-1β, IL-6, IL-17 protein ↓TNF-α protein ↓ICAM-1 protein ↓VCAM-1 protein ↑goblet cell density ↑tear production ↓corneal epithelium irregularities and desquamation	[146]
Catechin	Desiccation stress-induced NOD.B10.H2b mice	1% catechin or 11% hydrogen nanocomplex PEG/catechin as eye drops for 10 days	↓MMP-2 protein ↓MMP-9 protein ↓IL-1β, IL-6, IL-17 protein ↓TNF-α protein ↓ICAM-1 protein ↓VCAM-1 protein ↑goblet cell density	[148]
Ocular cancers				
Curcumin	C57/BL mice with the subretinal injection of melanoma B16/F10 cells	100 mg/kg intraperitoneally for 18 days	↓*MMP-2*, *MMP-9* mRNA and protein; ↓PI3K, EphA2 mRNA and protein; ↓tumor size	[202]
Glaucoma				
Resveratrol	Steroid-stimulated Sprague-Dawley rats	Topical, 0.2% twice-daily for 3 weeks	↑MMP-2 protein; ↑improvement of morphology of trabecular meshwork and retina; ↓IOP	[203]
Resveratrol	Steroid-stimulated Sprague-Dawley rats	Topical, 0.2% twice-daily for 1 week	↑MMP-2 protein; ↑uPA, tPA protein; ↓IOP	[204]
Theissenolactone C	Sprague-Dawley rats with normal saline injection into the anterior chamber	Single intraperitoneal injection of 10 mg/kg	↓MMP-9 protein and activity; ↓IL-1β, MCP-1 protein	[207]
Sodium 4-phenylbutyrate	Dexamethasone-stimulated C57BL/6J	1% sodium 4-phenylbutyrate as eye drops twice daily for 5 weeks	↓IOP; ↓fibronectin, collagen I protein	[216]

Legend: ↑ activation or increase; ↓ inhibition or decrease; EphA2—EPH Receptor A2; ICAM-1—intercellular adhesion molecule 1; IL—interleukin; iNOS—inducible nitric oxide synthase; IOP—intraocular pressure; MCP-1—monocyte chemoattractant protein 1; MMP-2—matrix metalloproteinase 2; MMP-9—matrix metalloproteinase 9; PI3K—phosphatidylinositol 3-kinase; TNF-α—tumor necrosis factor alpha; tPA—tissue plasminogen activator; uPA—urokinase plasminogen activator; VCAM-1—vascular cell adhesion protein 1; VEGF, vascular endothelial growth factor.

**Table 3 ijms-23-04256-t003:** Overview of clinical studies associated with the modulation of MMPs and TIMPs in eye diseases.

Agent	ClinicalTrials.gov Identifier/Phase (if Specified)	Participants/Enrollment	Dosage/Duration	Biological Effects/Findings	Reference
Dry eye syndrome					
Re-esterified omega-3 fatty acids	multicenter, prospective, interventional, placebo-controlled, double-masked study	105 patients with DES; the omega-3 group (n = 54) or placebo group (n = 51)	4 softgels daily with meals containing a total of either 1680 mg of eicosapentaenoic acid/560 mg of docosahexaenoic acid re-esterified omega-3 group or 3136 mg linoleic acid safflower oil as the control group for 12 weeks	↓tear MMP-9 ↓OSDI score ↑TBUT ↓tear film osmolarity	[212]
Re-esterified triglyceride form of omega-3 fatty acids	prospective comparative cohort study	66 patients complaining of new-onset non-specific typical dry eye 1 month after uncomplicated cataract surgery; the omega-3 group (n = 32) or placebo group (n = 34)	2 tablets containing a total of 1680 mg of eicosapentaenoic acid/506 mg of docosahexaenoic acid re-esterified triglyceride form of omega-3 two times per day for 2 months along with artificial teardrops	↓tear MMP-9 ↓OSDI score ↑tear production ↓Dry Eye Questionnaire score	[213]

Legend: ↑ activation or increase; ↓ inhibition or decrease; OSDI—ocular surface disease index; MMP-9—matrix metalloproteinase 9; TBUT—tear break-up time.

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
