# Peer review of "The Role of Metalloproteinases and Their Tissue Inhibitors on Ocular Diseases: Focusing on Potential Mechanisms"

_ijms, 2022, doi:10.3390/ijms23084256_

Round 1

Reviewer 1 Report

The manuscript by Caban and coworkers review the role played by metalloproteinases and proteinase inhibitors on eye diseases. The authors provide an interesting and updated perspective on this topic and summarize new modulators of expression and activity of these molecules with potential therapeutic activity in ophthalmology. The review represents a valuable source of information that will be of interest for people working in this field. However, some concerns need to be improved.

Specific points.

  1. In opinion of this reviewer, the main objective of the review, such as appears in the abstract section (line 16), must be revised. The role of MMPs in eye diseases cannot be “demonstrated” just by reviewing the literature.
  2. Many sentences need to be revised and clarified because some of them are confusing and others have missing words. The following are some examples: lines 8-10, 160-161, 264, 310-311, 399, 527 and 607.
  3. Line 364, change “healthy patients” to “control subjects”.
  4. Change “angle-closed” to “closed-angle”, lines 506 and 525.

Reviewer 2 Report

The manuscript seems well written and reviews the role of MMP's in eye diseases. However -and throughout the manuscript - the role of MMPs in diseases is vastly overestimated. For instance: line 171-line 174 "play essential roles is more actually something in the line of maybe associated with. Line 1999-line 205 "an important role should say something as implicated in skin melanoma as the manuscript describes a study basically using one cell line of an amelanotic melanoma known not to be an uveal melanoma. line 157 are pericytes characteristic of the tumor environment or just a component of the vasculature in general? I would suggest to not overestimate the role of MMPs too much throughout the manuscript and keep it at describing what the role actually is. (as it is also correctly done many times throughout the manuscript). .  
